# CARL: Constraint-Aware Reinforcement Learning for Planning with LLMs

## Abstract

Despite their strong reasoning capabilities and extensive world knowledge, Large Language Models (LLMs) often generate plans that ignore or violate task constraints, limiting their reliability in real-world planning scenarios. This stems from their limited ability to systematically incorporate constraint information during generation. Existing methods typically rely on external tools or task decomposition strategies, but do not improve the model's intrinsic awareness or understanding of constraints. To address this, we propose Constraint-Aware Reinforcement Learning (CARL), a novel training-based approach that systematically strengthens LLMs' intrinsic focus on constraints. CARL introduces a constraint-aware reward by comparing the model's output distributions under constrained and unconstrained inputs, encouraging constraint focus and penalizing neglect. The method is compatible with various RL frameworks and requires no external tools or top models. Extensive experiments on BlocksWorld, TravelPlanner, and T-Eval demonstrate that CARL outperforms standard RFT baselines and state-of-the-art reasoning models, and exhibits a markedly increased focus on constraint-related inputs. Our work enables scalable, end-to-end constraint-aware planning in LLMs, marking a step toward more autonomous and compliant language agents. Code and data will be released afterwards.

## 1 Introduction

Large Language Models (LLMs) have achieved impressive results in reasoning, tool use, and world knowledge modeling, making them strong candidates for complex planning tasks—an essential component of cognitive AI systems (Huang et al., 2022b; Ahn et al., 2022). Planning requires generating a sequence of executable actions to achieve a goal while strictly adhering to a set of constraints (Newell et al., 1958; Kartam & Wilkins, 1990). For example, a travel planning query might specify *who is traveling*, *when*, and *where*, with constraints such as *budget*, *transportation preferences*, or *dietary requirements*.

Despite their extensive world knowledge and strong reasoning abilities, LLMs consistently struggle to generate constraint-compliant plans in practice (Wei et al., 2025; Huang et al., 2024). On the real-world planning benchmark TravelPlanner, the advanced model DeepSeek-R1 (Guo et al., 2025), which performs strongly on general reasoning tasks, achieves a pass rate of only 12.2%, far below human-level performance. This significant gap is not due to weak reasoning ability but rather reflects a fundamental shortcoming: **LLMs lack the capacity to systematically incorporate constraints into their generation process**. Empirical studies (Xie et al., 2024b) support this view, showing that LLMs often neglect constraints during planning and consistently exhibit low attribution scores for constraint-related inputs.

Existing approaches attempt to circumvent this issue by offloading constraint reasoning to external tools or structured pipelines. Some adopt the *plan-then-execute* paradigm, decomposing complex queries into a sequence of simpler subtasks (Wang et al., 2023; Singh et al., 2022); others follow a *step-by-step* framework that interleaves planning with action execution in an iterative manner (Wei et al., 2022), or translate natural language queries into formal planning representations (e.g., PDDL) handled by symbolic solvers (Wu et al., 2022; He-Yueya et al., 2023). While these methods achieve improved performance through structured pipelines or external tools, they typically require closed-source models, external tools, or task-specific engineering. More importantly, they do not enhance

the model's intrinsic understanding of constraints, limiting generalizability and deployment in autonomous environments.

To address this, we propose **Constraint-Aware Reinforcement Learning (CARL)**, a general training framework designed to explicitly strengthen LLMs' focus on constraints through reinforcement learning. CARL introduces a novel constraint-aware reward, which captures how the model's output distribution shifts under constrained vs. unconstrained inputs. Specifically, we compute the KL divergence between log-probabilities in the two settings and use it as a reward signal, guiding the model to integrate constraint signals more explicitly. Compared to discrete task rewards, our continuous constraint-aware reward provides smoother optimization gradients while encouraging meaningful exploration during planning failures.

CARL is a novel training-based framework that directly enhances an LLM's intrinsic understanding and sensitivity to constraints. As illustrated in Figure 1, CARL-trained models exhibit significantly higher attribution scores for constraint tokens (e.g., non-self-driving transportation, Asian cuisine) than those trained with standard Reinforcement Fine-Tuning (RFT), leading to improved plan validity and execution. Furthermore, CARL is compatible with a broad range of RL algorithms, including both on-policy (e.g., PPO, GRPO) and off-policy (e.g., DPO) methods.

We evaluate CARL across three diverse planning benchmarks: BlocksWorld (block manipulation), TravelPlanner (travel planning), and T-Eval (tool use). Results show consistent and substantial improvements over standard Reinforcement Fine-Tuning (RFT), including a +11.1% gain and a 56.1% final pass rate on TravelPlanner—outperforming state-of-the-art reasoning LLMs like o1-preview (10.0%) and DeepSeek-R1 (12.2%). Ablation studies demonstrate the effectiveness of constraint-aware reward, while attribution analysis confirms that CARL successfully improves constraint awareness.

To summarize, our contributions are threefold:

- We propose CARL, a novel reinforcement learning framework that systematically enhances LLMs' focus on constraints by modeling distributional shifts in output log-probabilities under constrained versus unconstrained inputs. To the best of our knowledge, CARL is the first training-based work to explore a deeper integration of constraint-aware supervision signals beyond reward-level modifications.

- We design a learnable and interpretable constraint-aware reward mechanism that enables fine-grained control over constraint-compliant behavior through log-probability comparisons. This reward formulation is general and can seamlessly extend to both on-policy methods (e.g., PPO, GRPO) and off-policy paradigms such as DPO.

- We demonstrate CARL's effectiveness across diverse planning benchmarks, where it achieves substantial performance gains and significantly improved constraint focus, all without relying on external tools or top models.

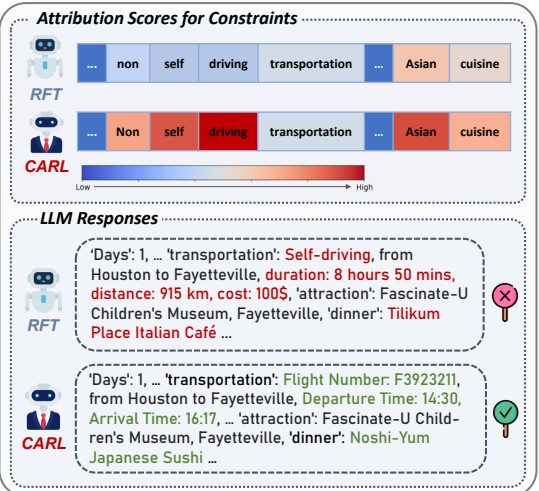

Figure 1: A typical case of planning tasks. The upper box demonstrates that a query can be decomposed into a goal and constraints, while the lower box shows that our CARL exhibits a higher focus on constraints, ultimately outperforming RFT in planning.

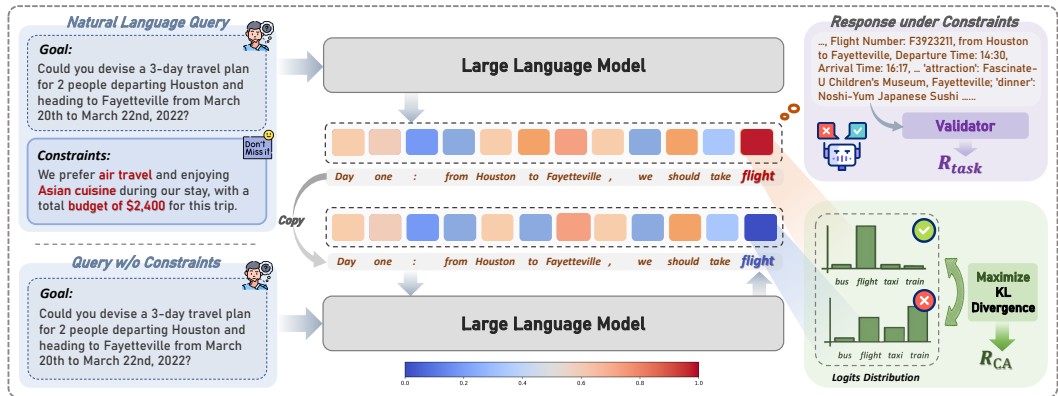

Figure 2: The framework of our proposed CARL. The reward is decomposed into two components: $R_{\text{task}}$ for achieving task-oriented objectives and $R_{\text{CA}}$ for sensitive adherence to constraints.

## 2 RELATED WORKS

### 2.1 PLANNING WITH LLMS

Large language models (LLMs) have demonstrated remarkable reasoning capabilities (Yao et al., 2023; Kojima et al., 2022; Raman et al., 2024) and effective tool utilization (Qin et al., 2023; Schick et al., 2023), positioning them as promising candidates for complex planning tasks. Leveraging LLMs' zero-shot generalization, numerous studies have explored direct zero-shot planning approaches (Huang et al., 2022a; Ahn et al., 2022). However, these methods are typically constrained to simplified scenarios and often require interactive step-by-step execution with environmental grounding. To address more complex planning challenges, researchers have adopted chain-of-thought (CoT) prompting to guide LLMs through structured reasoning processes (Wei et al., 2022). Recently, tool-augmented planning frameworks have emerged, enhancing LLM capabilities through external systems: some translate problems into formal representations for external planners (Liu et al., 2023; Xie et al., 2023; Gundawar et al., 2024), others integrate code snippets to handle dynamic "what-if" scenarios (Li et al., 2023), and several implement iterative refinement loops using task-specific verifiers or human feedback (Kambhampati et al., 2024; Chen et al., 2024).

The major commonalities of these approaches treat LLMs as static components—either as high-level task coordinators or natural language translators—rather than fundamentally enhancing their intrinsic planning capabilities. On this basis, they externalize complex reasoning and constraint handling, creating dependency on external tools and top models. While fine-tuning represents a direct pathway to improve model-intrinsic skills, its application to planning tasks remains surprisingly nascent, with limited research focusing on systematic constraint integration during plan generation.

Building on this observation, we propose a novel reinforcement learning paradigm that directly enhances the model's constraint-aware planning capabilities at the policy level. Unlike existing approaches that rely on external scaffolding, our method instills planning competence directly into the model through constraint-aware reward. This enables superior performance on complex planning tasks while achieving greater autonomy, without dependence on top models, external tools, or extensive prompt engineering.

### 2.2 REINFORCEMENT LEARNING FOR LLMS

Reinforcement Learning from Human Feedback (RLHF) (Kaufmann et al., 2024) adapts RL for LLMs by training them via the complex Proximal Policy Optimization (PPO) algorithm (Yu et al., 2022). To simplify this process, more efficient methods like Direct Preference Optimization (DPO) (Rafailov et al., 2023) and SimPO (Meng et al., 2024) were developed. However, these simpler methods often sacrifice performance and can suffer from off-policy issues. Recent approaches such as Group Relative Policy Optimization (GRPO) (Shao et al., 2024) and Reinforcement Learning

with Online Optimization (RLOO) (Ahmadian et al., 2024) continue to seek a balance between performance and efficiency.

Recent studies have enhanced models' input awareness through designed reward functions or constructed preference pairs. For instance, the attention scores are employed as reward signals in Kiruluta et al. (2025) to prioritize crucial tokens in inputs, while Gu et al. (2024); Deng et al. (2024) leverage preference pairs constructed from original and noise-perturbed images to strengthen focus on visual anchors. However, these approaches remain constrained to simple QA tasks, and their effectiveness in complex reasoning tasks such as planning remains unverified.

In this paper, we introduce a constraint-aware reward mechanism specifically designed for planning tasks. Our key innovation lies in quantifying constraint sensitivity through distributional shifts in log-probabilities between constrained and unconstrained conditions. This formulation is natively compatible with on-policy methods (PPO, GRPO), where it directly shapes the reward. We further demonstrate its seamless adaptation to off-policy frameworks like DPO by transforming constraint-aware signals into preference pairs that preserve the relative constraint compliance between responses.

## 3 PRELIMINARIES

### 3.1 PLANNING TASK FORMULATION AND DECOMPOSITION

To faithfully address the challenges in planning tasks, as shown in Figure 2, a planning task query can be decomposed into two components: goal (the final target to achieve) and constraints (the conditions that need to be adhered to). On this basis, we define an unconstrained planning problem to more systematically examine the role of constraints in achieving reliable planning outcomes. Our method is designed to be broadly applicable across generic planning scenarios. Specifically, a planning task $\mathcal{P}$ is represented as an input sequence:

$$x = (x_1, x_2, \ldots, x_T) \in \mathbb{R}^{T \times d} \tag{1}$$

where $T$ is the input length, and $d$ is the dimensionality of the embedding space.

Constraints are an inherent part of all planning tasks and play a pivotal role in ensuring outcome correctness. In our framework, such constraints are identified and modeled through the following procedures:

- For tasks with explicitly stated constraints, such as `TravelPlanner`, we utilize the predefined constraints explicitly described within the task query.
- For tasks with implicitly defined constraints, such as `T-Eval`, we employ constraint-extraction heuristics. Specifically, a lightweight prompt-based approach extracts and isolates the constraint-relevant portions of the input query, with full implementation details provided in Appendix A.

Formally, we denote the indices corresponding to the constraint tokens as $\mathcal{C} \subseteq \{1, 2, \ldots, T\}$. The extracted sequence of constraint-specific tokens is then $x_{\mathcal{C}} = (x_t)_{t \in \mathcal{C}}$. The remaining subsequence, which represents the unconstrained portion of the planning task, is defined as:

$$x_{\backslash \mathcal{C}} = (x_t)_{t \in \{1, \ldots, T\} \backslash \mathcal{C}} \tag{2}$$

While the unconstrained planning task $\mathcal{P}_{\backslash \mathcal{C}}$ may capture general goal-related elements of the task, it omits the crucial information encoded within constraints, potentially leading to incomplete or invalid solutions. This observation motivates our decomposition approach: constraints often act as the governing principles that disambiguate solutions and guarantee their feasibility. This distinction sets the stage for our proposed constraint-aware reinforcement learning framework.

### 3.2 GROUP RELATIVE POLICY OPTIMIZATION (GRPO)

GRPO (Shao et al., 2024) is an on-policy reinforcement learning algorithm. In the context of planning, consider a dataset $D$ containing datapoints consisting of inputs $x$. The GRPO learning objective with respect to the policy $\pi_\theta$ can be written as follows, where $\theta$ represents the parameters in a large language model:

$$\mathcal{J}_{\text{GRPO}}(\theta) = \mathbb{E}_{[\{y_i\}_{i=1}^G \sim \pi_{\theta_{old}}(Y|x)]} \frac{1}{G} \sum_{i=1}^{G} \frac{1}{|y_i|} \sum_{t=1}^{|y_i|} \Bigg\{$$

$$\min \left[ r_{i,t}(\theta)\hat{A}_{i,t}, \text{clip}\left(r_{i,t}(\theta), 1 - \epsilon_l, 1 + \epsilon_h\right)\hat{A}_{i,t} \right] - \beta \mathbb{D}_{KL}\left[\pi_\theta || \pi_{ref}\right] \Bigg\}$$

$$\text{with } r_{i,t}(\theta) = \frac{\pi_\theta(y_{i,t}|x, y_{i,<t})}{\pi_{\theta_{old}}(y_{i,t}|x, y_{i,<t})} \tag{3}$$

$G$ denotes the size of the group which contains multiple responses $Y$ sampled from the rollout policy $\pi_{\theta_{old}}$, corresponding to one input instance $x$. $\epsilon_l, \epsilon_h \in R$ are hyperparameters for clipping too large updates. The token-level advantage $\hat{A}_{i,t}$ is defined as the sequence-level reward normalized across the group.

## 4 METHODOLOGY

### 4.1 OVERVIEW

To address the dual objectives of goal attainment and constraint satisfaction in planning, we reformulate the standard reinforcement learning (RL) objective by decomposing the overall reward signal $R$ into two components, as shown in Figure 2: a task-specific reward $R_{\text{task}}$ for achieving task-oriented objectives, and a constraint-aware reward $R_{\text{CA}}$ for sensitive adherence to constraints:

$$R(x, y) = R_{\text{task}}(x, y) + \alpha R_{\text{CA}}(x, y) \tag{4}$$

where $\alpha \geq 0$ is a hyperparameter that regulates the relative importance of constraint adherence in the learning process. The trade-off between these components balances task achievement with constraint compliance.

At its core, our methodology embeds constraint-awareness into both the reward shaping and the policy optimization steps, detailed below.

### 4.2 CONSTRAINT-AWARE REWARD SHAPING

We introduce a novel reward shaping mechanism that explicitly integrates constraint-awareness into the learning objective. The task reward $R_{\text{task}}$ is computed as:

$$R_{\text{task}}(x, y) = \mathbb{I}[y \in \mathcal{Y}_{\text{valid}}(x)] \tag{5}$$

where $\mathbb{I}[\cdot]$ is an indicator evaluating to 1 if the generated output $y$ satisfies the task-specific success criterion (task achievement), and $\mathcal{Y}_{\text{valid}}(x)$ is the task-valid output space for input $x$. The constraint-aware reward $R_{\text{CA}}$ is defined as:

$$R_{\text{CA}}(x, y) = \mathbb{D}_{\text{KL}}[\pi_\theta(y|x) \| \pi_\theta(y|x_{\setminus C})] \tag{6}$$

Intuitively, the KL divergence captures the extent to which constraints influence model behavior, encouraging generation patterns that adhere to the constraint information. Combining this reward with the RL objective (e.g., GRPO) yields the complete CARL objective:

$$\mathcal{J}_{\text{CARL}}(\theta) = \mathbb{E}_{[\{y_i\}_{i=1}^G \sim \pi_{\theta_{old}}(Y|x)]} \frac{1}{G} \sum_{i=1}^{G} \frac{1}{|y_i|} \sum_{t=1}^{|y_i|} \Bigg\{$$

$$\min \left[ r_{i,t}(\theta)\hat{A}_{i,t}, \text{clip}\left(r_{i,t}(\theta), 1 - \epsilon_l, 1 + \epsilon_h\right)\hat{A}_{i,t} \right] - \beta \mathbb{D}_{KL}\left[\pi_\theta || \pi_{ref}\right]$$

$$+ \alpha \mathbb{D}_{\text{KL}}[\pi_\theta(y_i|x) \| \pi_\theta(y_i|x_{\setminus C})] \Bigg\} \tag{7}$$

where $i$ indexes the $i$-th rollout response. $\alpha$ and $\beta$ are weighting coefficients used for constraint-aware reward and KL penalty ($\mathbb{D}_{KL}[\pi_\theta || \pi_{ref}]$). We then compare the GRPO-version training dynamics and efficiency of CARL and RFT, as shown in Appendix B.

### 4.3 Constraint-Aware Direct Preference Optimization

To adapt our constraint-aware paradigm to preference-based learning, we propose a novel extension of Direct Preference Optimization (DPO) that injects constraint focus through strategic preference construction. Our key innovation lies in generating contrastive responses under constraint ablation to create informative preference pairs.

Given a standard response $y \sim \pi(\cdot|x)$ and its constraint-ablated counterpart $y_{\setminus \mathcal{C}} \sim \pi(\cdot|x_{\setminus \mathcal{C}})$, the preference dataset is defined as:

$$\mathcal{D}_{\text{CA}} = \left\{ (x^{(i)}, y^{(i)}, y_{\setminus \mathcal{C}}^{(i)}) \,|\, y^{(i)} \in \mathcal{Y}_{\text{valid}}(x^{(i)}) \right\}_{i=1}^{N} \tag{8}$$

The preference-based DPO objective is then augmented to reflect the impact of constraint ablation:

$$\mathcal{L}_{\text{CA-DPO}} = -\mathbb{E}_{\mathcal{D}_{\text{CA}}} \left[ \log \sigma \left( \beta \Delta(x, y, y_{\setminus \mathcal{C}}) \right) \right] \tag{9}$$

The constrained advantage function $\Delta$ is computed as:

$$\Delta(x, y, y_{\setminus \mathcal{C}}) = \log \frac{\pi_\theta(y|x)}{\pi_{\text{ref}}(y|x)} - \log \frac{\pi_\theta(y_{\setminus \mathcal{C}}|x)}{\pi_{\text{ref}}(y_{\setminus \mathcal{C}}|x)} \tag{10}$$

This formulation introduces an implicit reward margin that quantifies constraint influence on policy outputs, effectively incentivizing the model to generate constraint-compliant responses through optimization. The corresponding results are shown in Figure 3.

## 5 Experiments

### 5.1 Datasets and Settings

**Datasets.** We evaluate CARL across three complementary planning benchmarks that collectively cover *classical symbolic planning*, *real-world constrained decision-making*, and *tool-mediated planning*:

- **BlocksWorld** (Valmeekam et al., 2024) is a formal symbolic planning environment with well-defined action schemas and **static** constraints. Given an initial block configuration and a goal state, models must generate action sequences that strictly adhere to the physical constraints specified in the prompt. This benchmark provides a controlled setting to isolate and evaluate core constraint-handling capabilities.
- **TravelPlanner** (Xie et al., 2024a) presents a real-world travel planning challenge where models must generate plans based on provided information and user queries, aligning with commonsense and the hard constraints specified in the queries. Unlike the static nature of BlocksWorld, the hard constraints in TravelPlanner are **dynamic**, as they need to be inferred from the query and satisfied through item selection.
- **T-Eval** (Chen et al., 2023) is a fine-grained benchmark assessing LLMs' tool-use ability across multiple evaluation aspects. In this work, we primarily focus on its planning task. As noted in Sec. 3.1, T-Eval is characterized by **implicit** constraints, which are embedded within the input queries. We then use GPT-4o with a lightweight prompt to extract these constraints.

We follow the official partitions for BlocksWorld and TravelPlanner, using 100 and 45 samples for training, and 500 and 180 samples for testing, respectively. For T-Eval, we construct a training set by randomly selecting 128 samples from its 553-sample evaluation set.

**Metrics.** Accuracy is used for evaluating BlocksWorld, whereas precision, recall, and F1-score are used for T-Eval[1]. For TravelPlanner, we employ a multi-faceted evaluation framework that separates commonsense from hard constraints, with two complementary metrics reported for each:

- *Micro pass rate*: The ratio of successfully satisfied constraints to total constraints of that type.
- *Macro pass rate*: The ratio of plans satisfying all constraints of that type to total plans.

---

[1]In T-Eval, the reported precision, recall, and F1-score are the arithmetic means of the per-sample scores.

Table 1: Results on planning benchmarks. Unless specified, both RFT and CARL are implemented based on GRPO. The optimal results are in bold, and the suboptimal ones are underlined.

| Model | BlocksWorld | TravelPlanner | | | | | T-Eval | | |
| | | Commonsense | | Hard | | Final | Precision | Recall | F1-score |
| | | Micro | Macro | Micro | Macro | | | | |
| GPT-4o | 42.4 | 84.7 | 31.1 | 53.6 | 31.1 | 7.8 | 90.4 | 86.4 | 87.5 |
| o1-preview | 97.8 | 79.6 | 15.0 | 41.9 | 37.8 | 10.0 | 90.0 | 86.5 | 87.4 |
| DeepSeek-V3 | 44.8 | 80.3 | 17.2 | 30.5 | 13.9 | 2.2 | 91.1 | 87.4 | 88.5 |
| DeepSeek-R1 | 98.2 | 80.6 | 22.2 | 51.7 | 41.7 | 12.2 | 90.2 | 87.2 | 87.8 |
| Qwen2.5-72B-Instruct | 13.8 | 82.3 | 16.7 | 32.6 | 22.8 | 6.1 | 92.2 | 88.1 | 89.2 |
| QwQ-32B | 88.8 | 74.9 | 6.1 | 41.4 | 32.8 | 4.4 | 88.3 | 84.6 | 85.6 |
| Llama-3.1-70B-Instruct | 21.6 | 82.8 | 18.9 | 33.1 | 16.1 | 2.2 | 85.4 | 81.9 | 83.0 |
| DeepSeek-R1-Distill-Llama-8B | 1.4 | 61.2 | 0.0 | 0.0 | 0.0 | 0.0 | 81.8 | 79.3 | 79.4 |
| Qwen3-8B | 31.2 | 72.7 | 7.8 | 34.8 | 27.8 | 2.2 | 86.6 | 83.5 | 84.2 |
| DeepSeek-R1-Distill-Llama-8B (RFT) | 42.0 | 80.8 | 25.0 | 36.2 | 19.4 | 5.6 | 88.0 | 86.0 | 86.4 |
| DeepSeek-R1-Distill-Llama-8B (Ours) | 52.6 | 81.1 | 32.1 | 42.9 | 28.9 | 11.7 | 88.4 | **88.6** | _87.5_ |
| Qwen3-8B (RFT) | _73.8_ | _96.3_ | _74.4_ | _65.7_ | _48.9_ | _45.0_ | _88.6_ | 88.2 | 87.3 |
| Qwen3-8B (Ours) | **77.2** | **97.3** | **81.7** | **73.1** | **59.4** | **56.1** | **89.5** | _88.5_ | **88.1** |

Finally, we use the *final pass rate* as the proportion of plans satisfying all constraints, which corresponds to the *macro pass rate* when considering all constraints collectively, representing the ultimate planning success metric.

**Implementation Details.** Our reinforcement learning framework is implemented based on Verl (Sheng et al., 2025). Unless specified, we adopt Qwen3-8B as the base model to balance performance and training efficiency, and use GRPO for optimization. Training is conducted on a single node with 8 PPUs (comparable to A100 GPUs), and each step samples a batch of 64 queries with 8 rollouts. The weighting coefficients $\alpha$ and $\beta$ (see Equation 7) are both set to 0.001. Additional training details are provided in Appendix C.

## 5.2 CARL PERFORMANCE

**Results on Planning Benchmarks.** Table 1 summarizes the main evaluation results across all planning benchmarks, highlighting the comparative performance of CARL against state-of-the-art LLMs and RFT. There are three key takeaways:

First, CARL significantly boosts the planning capabilities of Qwen3-8B and DeepSeek-R1-Distill-Llama-8B. For example, with fewer than 128 training queries, CARL significantly boosts Qwen3-8B's planning performance, achieving up to 53.9% improvement. It consistently outperforms RFT—reaching a 56.1% pass rate on TravelPlanner—and significantly surpasses state-of-the-art reasoning-oriented models such as o1-preview (10.0%) and DeepSeek-R1 (12.2%). Furthermore, CARL enables the performance of the 14B model comparable to top models on BlocksWorld, as shown in Figure 4 and Table 11.

Second, CARL effectively mitigates constraint neglects or violations in multi-constraint planning tasks. While baseline models often perform well in terms of *micro pass rate* (i.e., the proportion of individual constraints satisfied), their *macro pass rate* (i.e., the proportion of plans satisfying all constraints) remains low. This indicates that performance bottlenecks arise from constraint neglect or violation in multi-constraint scenarios, rather than inherent difficulty in satisfying individual constraints. By enhancing the model's focus on constraints, CARL substantially mitigates such issues, leading to improved *macro* and *final pass rates*.

Last, CARL outperforms RFT in handling dynamic hard constraints. Although both CARL and RFT support the learning of static commonsense constraints, CARL proves more effective at handling dynamic constraints—hard constraints that vary across instances—resulting in higher overall task success rates than RFT.

To summarize, CARL consistently demonstrates superior planning performance and robustness across benchmarks and models. More results on planning benchmarks, including comparisons with SFT and agent-based methods, and generalization performance on other benchmarks, are presented in Appendix D.

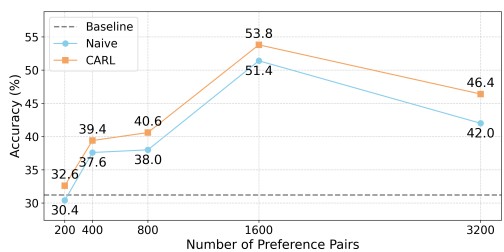

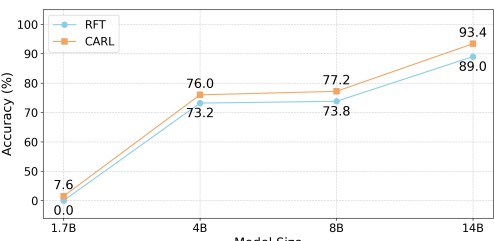

Figure 3: Performance comparison of naive DPO and CA-DPO with different preference pairs on BlocksWorld.

Figure 4: Performance comparison of RFT and CARL with different size Qwen3 models on BlocksWorld.

**Adaptation to Other RL Methods.** Our proposed CARL framework can be seamlessly extended to other representative on-policy and off-policy reinforcement learning methods, such as PPO and DPO. Table 2 summarizes the performance of these CARL implementations, consistently showing superior results compared to their naive counterparts.

The implementation of CARL on PPO is similar to that on GRPO; however, for DPO, we begin by constructing preference pairs using responses generated from the base model. Specifically, we input constrained queries and construct preference pairs based on the correctness of the responses for naive DPO. For CA-DPO, we input both constrained and unconstrained queries. The correct responses generated for constrained queries are selected as positive samples, while the responses generated for unconstrained queries are randomly chosen as negative samples, which are then utilized to construct preference pairs. For both methods, we maintain the same total number of preference pairs and use identical parameter settings for optimization. Experimental results, shown in Figure 3, reveal that CA-DPO consistently outperforms naive DPO. With as few as 200 preference pairs, CARL enables the model to be aware of constraints and slightly improves planning performance. As the number of preference pairs increases, the diversity of data expands, leading to a continuous improvement in the model's planning capabilities. However, when the number of preference pairs reaches 3,200, the model's performance begins to decline. We attribute this to a bottleneck in data diversity among the preference pairs generated by the base model. We think that introducing additional data sources in future work could further enhance the effectiveness of DPO. For more details about DPO, please refer to Appendix E.

Table 2: Performance comparison of RFT and CARL with different RL methods on BlocksWorld.

| Model | PPO | | DPO | |
|---|---|---|---|---|
| | Naive | CARL | Naive | CARL |
| Qwen3-8B | 78.4 | 81.6 | 51.4 | 53.8 |

### 5.3 ABLATION STUDIES

We demonstrate the effectiveness of our constraint-aware reward through ablation studies of model size, reward masking, and reward calculation strategies.

**Model Size.** To investigate the impact of model scale, we apply CARL to four Qwen3 models of increasing size. Results in Figure 4 highlight two key effects:

- **Exploration Guidance:** For small models lacking initial planning ability (e.g., Qwen3-1.7B), RFT fails due to the absence of task rewards during rollout. In contrast, CARL's constraint-aware reward provides smoother gradients, enabling effective exploration and gradual acquisition of planning skills through constraint focus.
- **Constraint Grounding:** For models with basic planning ability, CARL's advantage over RFT grows with scale, as larger models better leverage constraint signals to activate stronger reasoning patterns and boost planning performance.

**Reward Masking Strategy.** We assess the constraint-aware reward design by replacing the masked input from constraints to goals—another key element in planning. As shown in Table 3, this variant performs worse than RFT (None). We hypothesize that queries without constraints can still produce valid

Table 3: Ablation study of reward masking strategy on BlocksWorld.

| Model | None | Goal | Constraint |
|-------|------|------|------------|
| Qwen3-8B | 73.8 | 70.4 | 77.2 |

plans, allowing the reward to suppress constraint neglect. In contrast, removing goals often leads to incoherent outputs and noisy supervision.

**Reward Calculation Strategy.** To investigate the impact of the reward calculation strategy on training performance and dynamics, we measure the discrepancy between the log-probabilities in Equation 6 using four metrics: *difference* (dif), *absolute difference* (abs), *mean squared error* (mse), and *low-variance kl diver-*

Table 4: Ablation study of reward calculation strategy on BlocksWorld.

| Model | dif | abs | mse | low_var_kl |
|-------|-----|-----|-----|------------|
| Qwen3-8B | 76.6 | 76.0 | 0.2 | 77.2 |

*gence* (low_var_kl, which we used in our other experiments). As shown in Table 4, all strategies facilitate stable training, with the notable exception of mse, which leads to collapse. The impact of reward calculation strategy on training dynamics is shown in Appendix F.

## 5.4 ATTRIBUTION ANALYSIS

To understand the source of planning performance improvements, we conduct an attribution analysis on BlocksWorld and TravelPlanner. We compare the base model with versions fine-tuned using RFT and CARL, focusing on the mean and distribution of attribution scores for constraint-related inputs (all scores are normalized by the response length). As shown in Figure 5, better task performance consistently correlates with higher attribution scores. Compared to RFT, CARL further enhances the model's focus on constraints, leading to stronger performance. Importantly, this improvement is not driven by a few outliers but reflects a consistent shift in the overall distribution, indicating that CARL's gains stem from a general enhancement in constraint sensitivity. Additional details and the case study are provided in Appendix G.

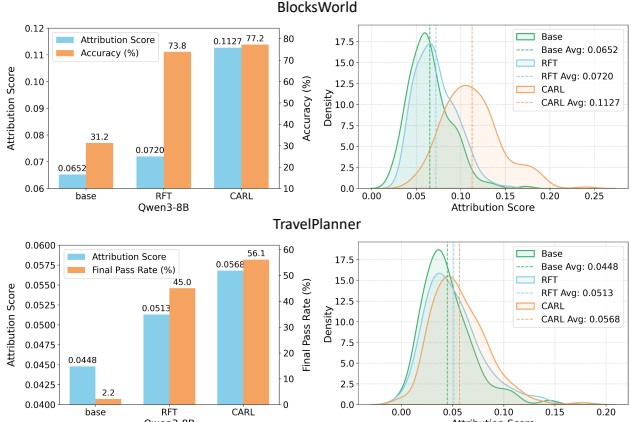

Figure 5: Attribution analysis on BlocksWorld and TravelPlanner. The left simultaneously presents the average attribution scores and performance of different models, while the right illustrates score distributions through Kernel Density Estimation.

## 6 CONCLUSION

We introduced CARL, a novel training-based framework that enhances LLMs' intrinsic sensitivity to constraints for planning tasks. By comparing model outputs under constrained and unconstrained inputs, CARL provides effective reward signals to improve constraint compliance without relying on external tools or task-specific engineering. Experiments on diverse benchmarks show that CARL outperforms both standard RFT and SOTA reasoning models, with improved attribution to constraints and strong generalization across RL methods and model scales. CARL demonstrates the promise of learning-based strategies for developing autonomous and constraint-aware language agents. Details regarding our use of Large Language Models (LLMs) are provided in Appendix H.

ETHICS STATEMENT

In accordance with the ICLR Code of Ethics, we address the ethical considerations of our work. Our research is conducted on publicly available academic benchmarks (e.g., BlocksWorld, TravelPlanner), and we did not involve human subjects. To ensure reproducibility, we will release our source code. We acknowledge two primary ethical risks. First, as CARL fine-tunes pre-trained models, it inherits but does not mitigate their intrinsic societal biases (e.g., in travel recommendations), which could lead to inequitable outputs. Second, improving the planning capabilities of LLMs presents a dual-use risk; we caution that the deployment of such powerful agents in high-stakes, real-world scenarios requires significant safety validation and human oversight. The authors declare no competing interests.

REPRODUCIBILITY STATEMENT

The source code, including main scripts required to reproduce our experiments, is available in the supplementary materials and will be released publicly upon publication. The datasets used are publicly available, and we describe our data preprocessing prompt in Appendix A. Frameworks and hyperparameters for our experiments are detailed in Appendix C, E, and G.

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

## A PROMPT USED FOR CONSTRAINT-EXTRACTION

We herein present the prompt designed to extract implicit constraints in T-Eval, as referenced in Sec. 3.1, along with its effect on T-Eval.

**Full Prompt.** We use a simple prompt to extract and isolate the constraint-relevant portions of the input query for T-Eval, as shown below:

```
You are an expert at simplifying user queries by removing specific
constraints while preserving the core intent.

Your task is to remove detailed constraints and specific requirements
from the user query, keeping only the main objectives and essential
context.

Rules:
1. Keep the main purpose and core actions
2. Remove specific numbers, quantities, limits
3. Remove specific dates, times, or temporal constraints
4. Remove detailed specifications or precise requirements
5. Maintain the overall structure and flow of the original query
6. Keep professional context and role descriptions
7. The output should be a simplified version that captures the essence
without the fine-grained constraints

Now process the following:
Original user query:
{user_prompt}

Simplified query (remove constraints but keep core intent):
```

**Effect on T-Eval.** We use two cases to demonstrate the effect of our prompt on T-Eval, with the extracted implicit constraints underlined, as shown below:

```
[Case 1]:

Constrained Queries:
As the office manager, I need to find a meeting room
that is available for the next two hours for a team meeting today. Once
an available room is found, please book it for the specified duration,
starting from the current time.

Unconstrained Queries:
As the office manager, I need to find a meeting room for a team meeting
and book it.

[Case 2]:

Constrained Queries:
I am writing a research paper on quantum computing, and I need
information about the first author of the articles. Please find articles
related to quantum computing and provide me with the meta information of
the first three articles. Lastly, I need to know if there are any meeting
 rooms available tomorrow from 2:00 PM to 4:00 PM.

Unconstrained Queries:
I am writing a research paper on quantum computing, and I need
information about the author of the articles. Please find articles
related to quantum computing and provide me with the meta information of
the articles. Lastly, I need to know if there are any meeting rooms
available tomorrow.
```

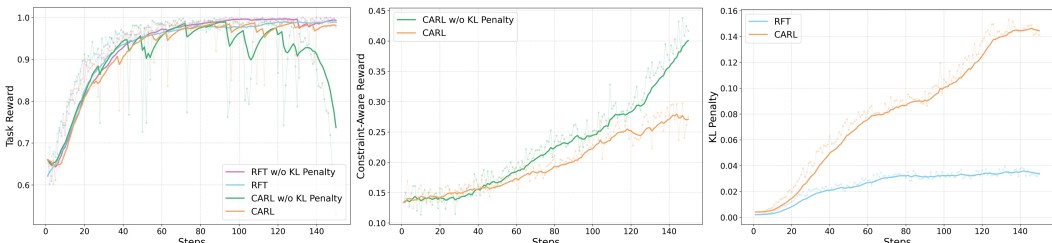

Figure 6: Comparison of the training dynamics on the task reward, constraint-aware reward and KL penalty. Solid lines indicate exponential moving averages of the data.

## B  COMPARISON WITH RFT

We herein compare the GRPO-version training dynamics and efficiency of CARL and RFT.

**Training Dynamics.** Since we maximize a KL divergence that is theoretically unbounded, the model may "hack" constraint-aware reward, eventually leading to performance collapse. The KL penalty, however, can mitigate the risk of reward hacking by constraining the magnitude of policy updates. Therefore, we investigate the impact of the KL penalty on the training dynamics of CARL and compare it with RFT. As shown in Figure 6, for RFT (blue line), the pure task reward does not introduce significant hacking risks, causing the KL penalty to remain relatively stable throughout training. Consequently, RFT can converge even without the KL penalty (purple line). In contrast, for CARL (orange line), the KL penalty increases in tandem with the constraint-aware reward during training, which prevents the model from reward hacking and subsequent collapse. Conversely, without the constraint of the KL penalty, the model begins to exploit the constraint-aware reward, ultimately leading to a collapse in performance (green line). Table 5 summarizes the results of the four training methods (RFT$_{\setminus Penalty}$ and CARL$_{\setminus Penalty}$ denote the versions of RFT and CARL without the KL penalty, respectively).

Table 5: Performance comparison of four training methods on BlocksWorld.

| Model | RFT$_{\setminus Penalty}$ | RFT | CARL$_{\setminus Penalty}$ | CARL |
|---|---|---|---|---|
| Qwen3-8B | 74.4 | 73.8 | 43.4 | 77.2 |

**Training Efficiency.** To evaluate training efficiency, we measure the average time per training step (in seconds) and planning accuracy across different settings (Qwen3-8B is used here). As shown in Table 6, increasing rollouts from 8 to 16 modestly improves RFT but increases about 70% more computational cost. In contrast, CARL achieves better performance with only a 10% overhead, striking a better balance between efficiency and effectiveness. This advantage is due to CARL's design: although it generates an additional set of unconstrained queries, these require only a single forward pass for log-probabilities and are excluded from the expensive rollout process.

Table 6: Efficiency analysis with different *Rollouts*, *Time per Training Step* and *Accuracy* on BlocksWorld.

| Method | Rollouts | Time per Step | Accuracy |
|---|---|---|---|
| RFT | 8 | 406.2 | 73.8 |
|  | 16 | 674.2 | 75.0 |
| CARL | 8 | 445.7 | 77.2 |

## C  IMPLEMENTATION DETAILS FOR GRPO

We herein present more details about benchmarks, training, and inference for GRPO.

**Benchmarks.** For BlocksWorld, we employ the official one-shot setting. For TravelPlanner, we adopt the "sole-planning" mode to focus on the LLMs' planning ability, excluding the influence of information gathering abilities required in the "two-stage" mode. For T-Eval, we use the default setting.

Table 7: Performance comparison with SFT on planning benchmarks. The optimal results are in bold, and the suboptimal ones are underlined.

| Method | BlocksWorld | TravelPlanner | | | | | T-Eval | | |
| | | Commonsense | | Hard | | Final | Precision | Recall | F1-score |
| | | Micro | Macro | Micro | Macro | | | | |
| --- | --- | --- | --- | --- | --- | --- | --- | --- | --- |
| DeepSeek-R1-Distill-Llama-8B | 1.4 | 61.2 | 0.0 | 0.0 | 0.0 | 0.0 | 81.8 | 79.3 | 79.4 |
| +SFT | 12.6 | 78.1 | 16.7 | 29.8 | 16.1 | 3.9 | 87.1 | 86.5 | 86.0 |
| +RFT | 42.0 | 80.8 | 25.0 | 36.2 | 19.4 | 5.6 | 88.0 | 86.0 | 86.4 |
| +CARL (Ours) | **52.6** | **81.1** | **32.1** | **42.9** | **28.9** | **11.7** | **88.4** | **88.6** | **87.5** |
| Qwen3-8B | 31.2 | 72.7 | 7.8 | 34.8 | 27.8 | 2.2 | 86.6 | 83.5 | 84.2 |
| +SFT | 14.2 | 77.4 | 25.0 | 24.3 | 14.4 | 8.3 | 84.0 | 84.3 | 83.5 |
| +RFT | 73.8 | 96.3 | 74.4 | 65.7 | 48.9 | 45.0 | 88.6 | 88.2 | 87.3 |
| +CARL (Ours) | **77.2** | **97.3** | **81.7** | **73.1** | **59.4** | **56.1** | **89.5** | **88.5** | **88.1** |

**Training.**  For GRPO, we employ the Verl framework. Regarding hyperparameters, we set the policy LLM learning rate to 1e-6 and sample 8 responses per query. Training is conducted on a single node with 8 PPUs (comparable to A100 GPUs), with a total batch size of 64. The maximum response length is set to 8,192 tokens. To optimize GPU memory usage, we enable gradient checkpointing and use Fully Sharded Data Parallel (FSDP) with CPU offloading.

For efficient LLM rollouts, we adopt vllm with a tensor parallel size of 1 and GPU memory utilization ratio of 0.6. The rollout sampling uses a temperature of 0.6 and a top-p value of 1.0. The weighting coefficients $\alpha$ and $\beta$ (see Equation 7) are both set to 0.001.

We train the model for 150 steps on BlocksWorld and T-Eval, and 300 steps on the more challenging TravelPlanner. The task reward is computed using the evaluation script provided by each benchmark.

**Inference.**  During inference, we employ vllm with a sampling temperature of 0.6 and set the maximum response length uniformly to 30,000 tokens to prevent truncation.

## D  MORE RESULTS ON PLANNING BENCHMARKS

We herein present more results on planning benchmarks, including comparisons with SFT and agent-based methods, generalization performance on other planning benchmarks, and more results on BlocksWorld.

### D.1  COMPARISON WITH SFT

Table 7 shows the performance comparison with SFT on planning benchmarks.

**Analysis.**  The experimental results demonstrate that RL-based methods are more effective than SFT at improving planning capabilities. We attribute this to two primary reasons. First, the ground-truth data provided by these benchmarks often contains only the final answer without the intermediate reasoning process. Consequently, direct SFT is not conducive to the model's ability to "think" and learn high-quality planning patterns. Second, planning tasks may have multiple feasible solutions that satisfy the given constraints, yet the ground truth typically provides only one. SFT may thus cause the model to memorize this specific ground-truth solution rather than genuinely learning how to plan.

### D.2  COMPARISON WITH AGENT-BASED METHODS

We herein introduce three agent-based methods for planning performance comparison:

• LLM-Modulo (ICML 2024): A planning agent based on feedback from external constraint critics from Kambhampati et al. (2024).

Table 8: Performance comparison with agent-based methods on TravelPlanner.

| Method | Base Model | External Planner | TravelPlanner | | | | |
| --- | --- | --- | --- | --- | --- | --- | --- |
| | | | Commonsense | | Hard | | Final |
| | | | Micro | Macro | Micro | Macro | |
| LLM-Modulo Multi-Agent | GPT-4 | No | - | - | - | - | 20.0 |
| | | | 90.0 | 41.7 | 55.7 | 48.3 | 31.7 |
| SMT Solver | GPT-4 Mistral-Large | Yes | 95.0 | 95.0 | 95.7 | 98.9 | 93.3 |
| | | | 72.0 | 70.6 | 63.3 | 66.7 | 66.7 |
| CARL (Ours) | Qwen3-8B | No | 97.3 | 81.7 | 73.1 | 59.4 | 56.1 |

- Multi-Agent (COLING 2025): A collaborative multi-agent system for planning based on task decomposition from Zhang et al. (2024).
- SMT Solver (NAACL 2025): A planning agent based on results of an external optimization solver from Hao et al. (2024).

**Results.**  We chose the complex real-world planning task TravelPlanner for comparison. Notably, TravelPlanner operates in two modes: "sole-planning" and "two-stage". The former focuses on LLMs' planning ability, with all relevant travel information provided as a lengthy textual context within the prompt, while the latter requires LLMs to gather information and complete planning. The aforementioned agent-based methods are specifically designed with tailored prompts for the "two-stage" mode and cannot be directly adapted to the "sole-planning" mode (which we use in the main results). We therefore report their performance under the "two-stage" mode, with results shown in Table 8.

**Analysis.**  The experimental results demonstrate that while current agent-based methods (e.g., LLM-Modulo and Multi-Agent system) exhibit strong performance in complex planning tasks, their effectiveness typically relies on top models like GPT-4 combined with external tools (e.g., constraint critics) or multi-agent collaboration mechanisms to complete reasoning and planning processes. In contrast, our proposed CARL framework differs fundamentally in design philosophy: it neither depends on top models nor introduces external tools or multi-agent architectures, but instead achieves end-to-end scalable constraint-aware planning within a single lightweight model.

Notably, using only the Qwen3-8B model – a medium-scale model – CARL still outperforms two of the three agent-based baselines (LLM-Modulo and Multi-Agent). This indicates that by internalizing constraint comprehension within the model's planning process, CARL effectively compensates for its scale limitations while demonstrating robust planning capabilities.

Although the SMT solver-based method currently achieves state-of-the-art performance, its superiority heavily depends on a tightly-coupled GPT-4 + external planner architecture. We observe significant performance degradation when replacing GPT-4 with the equally capable 123B Mistral-Large model, revealing excessive dependency on specific architectures (particularly GPT-4). Furthermore, this method requires external planners to search through vast solution spaces, incurring substantial latency (averaging 245 seconds per query on TravelPlanner) – over 24× slower than CARL's inference efficiency (10s average). Such high latency and strong dependencies limit its practical scalability and deployment flexibility.

Given the fundamental differences between CARL and existing agent-based methods in system architecture, model dependency, planning workflow, and external tool usage, direct "end-to-end" performance comparisons may lack fairness and interpretability due to inconsistent experimental setups. We therefore exclude direct performance comparisons with these methods in the main results to ensure experimental rigor and comparability.

Nevertheless, it must be emphasized that CARL achieves planning capabilities comparable to or even surpassing those of GPT-4-based agent methods with complex engineering architectures – all without external solvers, top models, or exceeding 8B parameters. This breakthrough highlights

Table 10: Generalization performance from TravelPlanner to TripCraft.

| Method | Category | TripCraft | | | | |
| | | Commonsense | | Hard | | Final |
| | | Micro | Macro | Micro | Macro | |
|---|---|---|---|---|---|---|
| Qwen3-8B | 3-day | 90.8 | 0.0 | 17.3 | 14.8 | 0.0 |
| | 5-day | 71.6 | 0.0 | 0.4 | 0.4 | 0.0 |
| | 7-day | 65.8 | 0.0 | 0.0 | 0.0 | 0.0 |
| +RFT | 3-day | 90.1 | 1.8 | 23.2 | 21.3 | 0.2 |
| | 5-day | 72.3 | 0.0 | 0.7 | 0.4 | 0.0 |
| | 7-day | 71.8 | 0.0 | 0.0 | 0.0 | 0.0 |
| +CARL | 3-day | 94.8 | 2.9 | 27.3 | 25.7 | 1.1 |
| | 5-day | 72.7 | 0.5 | 1.3 | 1.3 | 0.0 |
| | 7-day | 72.3 | 0.0 | 0.0 | 0.0 | 0.0 |

CARL's significant potential in empowering medium- and small-scale language models to achieve autonomous, efficient, and compliant planning, while establishing a new technical pathway for developing independent, lightweight, and reliable intelligent agents.

### D.3 GENERALIZATION PERFORMANCE ON OTHER PLANNING BENCHMARKS

We herein introduce two additional benchmarks to verify the generalization ability of CARL:

- Mystery BlocksWorld (NeurIPS 2023): A planning benchmark (Valmeekam et al., 2023) that is semantically identical to the standard BlocksWorld but syntactically obfuscated. All meaningful identifiers for objects (e.g., block), predicates (e.g., on-table), and actions (e.g., pickup) are replaced with misleading terms (e.g., harmony, attack, feast). This design eliminates any reliance on commonsense knowledge embedded in language, creating a rigorous generalization test that is particularly suited for models trained on the BlocksWorld dataset.
- TripCraft (ACL 2025): A real-world travel planning benchmark (Chaudhuri et al., 2025) designed to address the critical limitations of its predecessors, such as TravelPlanner. While TravelPlanner relies on semi-synthetic data, leading to significant spatial inconsistencies (e.g., assigning accommodations for a New York trip across 312 different cities), TripCraft is constructed entirely from real-world data sources to ensure geographical coherence and realistic itinerary generation. Furthermore, TripCraft significantly enhances task complexity by integrating crucial real-world constraints that are absent or underdeveloped in TravelPlanner, including public transit schedules, event availability, fine-grained attraction categories, and more nuanced user personas that capture travel styles and preferences. This makes TripCraft a more challenging and realistic benchmark, ideal for evaluating the generalization capabilities of models trained on the TravelPlanner dataset.

**Results.** We evaluate the generalization performance of three models—the Qwen3-8B base, and variants fine-tuned with RFT and CARL—on two distinct benchmarks. For the Mystery BlocksWorld evaluation, the variants were trained on BlocksWorld, and we use the official one-shot setting. In contrast, for the TripCraft evaluation, they were trained on TravelPlanner, and we adopt the "w/o Parameter Info" setting as per the TravelPlanner protocol. The results are summarized in Table 9 and Table 10, respectively.

Table 9: Generalization performance from BlocksWorld to Mystery BlocksWorld.

| Model | Base | RFT | CARL |
|---|---|---|---|
| Qwen3-8B | 0 | 0.7 | 1.8 |

**Analysis.** The results from both generalization experiments consistently demonstrate the superior generalization capabilities of CARL compared to the base model and RFT.

On the Mystery BlocksWorld benchmark (Table 9), which tests pure logical generalization by obfuscating syntax, the base model completely fails with a score of 0. While RFT shows a marginal improvement (0.7), CARL achieves a higher score of 1.8. This indicates that CARL is more effective at learning the underlying logical structure of the planning task, rather than merely memorizing the surface-level syntax of the training data.

Similarly, when generalizing from TravelPlanner to the more complex and realistic TripCraft benchmark (Table 10), CARL consistently outperforms its counterparts. This is particularly evident in the 3-day scenario, where CARL achieves the highest scores across all metrics. Most notably, it obtains a Final pass rate of 1.1, whereas the base model scores 0 and RFT only reaches 0.2. Furthermore, CARL demonstrates a stronger ability to satisfy diverse constraints, as shown by its superior Macro pass rates for both Commonsense (2.9) and Hard (25.7) constraints, where other methods struggle.

### D.4 MORE RESULTS ON BLOCKSWORLD

We apply CARL to other DeepSeek-R1-Distill models and compare it against RFT. As shown in Table 11, CARL consistently outperforms RFT. These results, combined with the performance of DeepSeek-R1-Distill-Llama-8B in Table 1, suggest that CARL can generalize to different model architectures. Notably, CARL achieves performance on par with top models at the 14B scale on BlocksWorld.

Table 11: More results with DeepSeek-R1-Distill models on BlocksWorld.

| Method | Qwen-7B | Qwen-14B |
|--------|---------|----------|
| RFT    | 24.6    | 86.0     |
| CARL   | 29.8    | 93.0     |

## E IMPLEMENTATION DETAILS FOR DPO

We herein provide more details about preference pair construction and training for DPO.

**Preference Pair Construction.** For Naive-DPO, we input constrained queries and construct preference pairs based on the correctness of the responses. For CA-DPO, we input both constrained and unconstrained queries. The correct responses generated for constrained queries are selected as positive samples, while the responses generated for unconstrained queries are randomly chosen as negative samples, which are then utilized to construct preference pairs. Our sampling strategies for preference pair construction can be summarized as shown in Table 12.

**Training.** For DPO, we employ the LLaMA-Factory framework. We consistently train the model for 3 epochs with a batch size of 64, a

Table 12: The sampling strategies used in Naive-DPO and CA-DPO. In CA-DPO, the rejected samples are outputs generated under conditions where constraints are absent, represented as $y^{(i)}_{\setminus \mathcal{C}}$. This design choice aims to maximize the divergence between $y^{(i)}_{\setminus \mathcal{C}}$ and the valid constrained output $y^{(i)}$. By doing so, the model is better guided to understand and incorporate the crucial role played by constraints in its learning process.

| Sampling Strategy | Chosen Samples | Rejected Samples |
|-------------------|----------------|------------------|
| Naive-DPO | $y^{(i)} \in \mathcal{Y}_{\text{valid}}(x^{(i)})$ | $y^{(i)} \notin \mathcal{Y}_{\text{valid}}(x^{(i)})$ |
| CA-DPO | $y^{(i)} \in \mathcal{Y}_{\text{valid}}(x^{(i)})$ | $y^{(i)}_{\setminus \mathcal{C}}$ |

learning rate of 2.0e-6, a preference coefficient $\beta$ of 0.1, and a warmup ratio of 0.05 on BlocksWorld.

## F IMPACT OF REWARD CALCULATION STRATEGY ON TRAINING DYNAMICS

The results from the ablation study, presented in Table 4, clearly show that the choice of reward calculation metric has a profound impact on training outcomes. While the dif, abs, and low_var_kl strategies all yield high and comparable final performance (76.6, 76.0, and 77.2, respectively), the mse strategy leads to a complete training collapse, with the model achieving a near-zero score of 0.2.

The training dynamics, illustrated in Figure 7, reveal the underlying reasons for this disparity. The mse strategy (green line) initially shows a promising increase in task reward, but it abruptly collapses

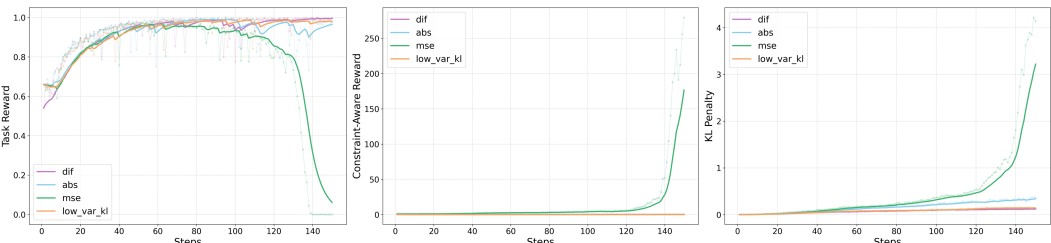

Figure 7: Impact of reward calculation strategy on training dynamics. Solid lines indicate exponential moving averages of the data.

after approximately 120 steps. This collapse is directly correlated with an explosive, uncontrolled spike in both the constraint-aware reward and the KL penalty, indicating severe training instability. We hypothesize that the quadratic nature of mse creates a volatile reward signal, excessively penalizing larger deviations and leading to policy divergence.

In contrast, dif, abs, and low_var_kl all maintain a stable and high task reward throughout training. Their constraint-aware rewards remain close to zero, providing a consistent and stable learning signal. Notably, low_var_kl and dif also maintain the lowest KL penalty, suggesting they guide the policy towards improvement more efficiently without deviating drastically from the reference model. This combination of high final performance and superior training stability confirms that low_var_kl is the most robust and effective reward calculation strategy for our framework.

## G  DETAILS FOR ATTRIBUTION ANALYSIS

**Setup.**  In this paper, we adopt Permutation Feature Importance (FeatureAblation in captum) as our analysis strategy for testing the inner workings of LLMs when planning. For TravelPlanner, we conduct analysis using the entire validation set (180 samples). For BlocksWorld, we randomly select 200 samples from the test set due to the high computational cost.

**Case Study.**  We present several cases used in our attribution analysis. The model after RFT fails in planning due to its insufficient sensitivity to constraints, whereas the model after CARL succeeds in planning by adequately attending to the constraints. Due to the length of the full response, we present only the portion following the "</think>" identifier.

The case on BlocksWorld demonstrates that the model after RFT executes a pick-up action immediately following an unstack operation, violating the fundamental constraint that prohibits pick-up or unstack operations while holding a block. The related constraints are explicitly highlighted with underlines within the case.

The case on TravelPlanner demonstrates that the model after CARL proactively considers whether daily accommodation and meal arrangements align with user preferences, thus producing plans that fully satisfy all constraints. This focus on preferences is partially explicitly highlighted with double underscores within the case.

```
1026

1027   [BlocksWorld]

1028
1029   Query:
       I am playing with a set of blocks where I need to arrange the blocks into
1030    stacks. Here are the actions I can do

1031
1032   Pick up a block
       Unstack a block from on top of another block
1033   Put down a block
1034   Stack a block on top of another block

1035
1036   I have the following restrictions on my actions:
1037   I can only pick up or unstack one block at a time.
       I can only pick up or unstack a block if my hand is empty.
1038   I can only pick up a block if the block is on the table and the block is
1039   clear. A block is clear if the block has no other blocks on top of it and
1040    if the block is not picked up.
1041   I can only unstack a block from on top of another block if the block I am
1042    unstacking was really on top of the other block.
1043   I can only unstack a block from on top of another block if the block I am
1044    unstacking is clear.
       Once I pick up or unstack a block, I am holding the block.
1045   I can only put down a block that I am holding.
1046   I can only stack a block on top of another block if I am holding the
1047   block being stacked.
1048   I can only stack a block on top of another block if the block onto which
1049   I am stacking the block is clear.
1050   Once I put down or stack a block, my hand becomes empty.
       Once you stack a block on top of a second block, the second block is no
1051   longer clear.

1052
1053   [STATEMENT]
1054   As initial conditions I have that, the blue block is clear, the hand is
       empty, the red block is on top of the orange block, the blue block is on
1055   top of the yellow block, the yellow block is on top of the red block and
1056   the orange block is on the table.
1057   My goal is to have that the blue block is on top of the red block and the
1058    yellow block is on top of the orange block.

1059
1060   My plan is as follows:

1061   [PLAN]
1062   unstack the blue block from on top of the yellow block
1063   put down the blue block
       unstack the yellow block from on top of the red block
1064   stack the yellow block on top of the blue block
       unstack the red block from on top of the orange block
1065   put down the red block
1066   unstack the yellow block from on top of the blue block
1067   stack the yellow block on top of the orange block
1068   pick up the blue block
       stack the blue block on top of the red block
1069   [PLAN END]
1070
1071
1072   [STATEMENT]
1073   As initial conditions I have that, the red block is clear, the orange
       block is clear, the hand is empty, the red block is on top of the yellow
1074   block, the orange block is on top of the blue block, the blue block is on
1075    the table and the yellow block is on the table.
1076   My goal is to have that the blue block is on top of the yellow block and
1077   the yellow block is on top of the red block.
1078
1079   My plan is as follows:
```

```
[PLAN]

RFT Response:
The user's plan contains errors due to misalignment with the initial
conditions. Here's the corrected plan:

Correct Plan:

1. Unstack the red block from the yellow block
   - Red is on top of yellow; red is clear (no blocks above it).
   - Unstack red and place it on the table.

2. Pick up the yellow block
   - Yellow is now on the table (after red is removed) and is clear.

3. Stack the yellow block on the red block
   - Red is on the table (clear), so yellow can be stacked on it.
   - Result: Red → Yellow.

4. Unstack the orange block from the blue block
   - Orange is on top of blue; orange is clear (no blocks above it).
   - Unstack orange and place it on the table.

5. Pick up the blue block
   - Blue is now on the table (after orange is removed) and is clear.

6. Stack the blue block on the yellow block
   - Yellow is on top of red and is clear (no blocks above it).
   - Result: Red → Yellow → Blue.

Final Arrangement:
- Red block is on the table.
- Yellow block is on top of red.
- Blue block is on top of yellow.

This sequence adheres to all constraints and achieves the goal. The user'
s original plan incorrectly assumed blue was on yellow initially, which
was not the case.

CARL Response:
To achieve the goal of having blue on top of yellow and yellow on top of
red, given the initial configuration:

- Red is on top of yellow
- Orange is on top of blue
- Blue and yellow are on the table
- Red and orange are clear

We need to rearrange the blocks while adhering to the rules of block
manipulation. Here's the step-by-step plan:

---

Plan

1. Unstack the red block from on top of the yellow block
   - Red is clear (initial condition), and it is on top of yellow.
   - After unstacking, red is held in the hand.
   - Yellow is now clear (since red is removed from above it).

2. Put down the red block
   - Hand becomes empty.
   - Red is now on the table.

3. Pick up the yellow block
```

```
    - Yellow is now clear (no blocks on top).
    - Hand is empty, so this is allowed.

4. Stack the yellow block on top of the red block
    - You are holding yellow, and red is clear (before stacking).
    - After stacking, red is no longer clear (rule: stacking makes the
    base block non-clear).
    - Hand becomes empty.

5. Unstack the orange block from on top of the blue block
    - Orange is clear (initial condition), and it is on top of blue.
    - After unstacking, orange is held in the hand.
    - Blue is now clear (since orange is removed from above it).

6. Put down the orange block
    - Hand becomes empty.
    - Orange is now on the table.

7. Pick up the blue block
    - Blue is now clear (no blocks on top).
    - Hand is empty, so this is allowed.

8. Stack the blue block on top of the yellow block
    - You are holding blue, and yellow is clear (no blocks on top).
    - After stacking, yellow is no longer clear (rule: stacking makes the
    base block non-clear).
    - Hand becomes empty.

---

Final Arrangement

- Red block is on the table.
- Yellow block is on top of red.
- Blue block is on top of yellow.

This satisfies the goal: blue is on top of yellow, and yellow is on top
of red.

---

Answer
```
[PLAN]
unstack the red block from on top of the yellow block
put down the red block
pick up the yellow block
stack the yellow block on top of the red block
unstack the orange block from on top of the blue block
put down the orange block
pick up the blue block
stack the blue block on top of the yellow block
[PLAN END]
```
```

```
[TravelPlanner]

Query:
I'm looking for a 7-day travel itinerary for 2 people, starting from Reno
 and heading to Texas, specifically visiting 3 different cities. The
travel dates are from March 7th to March 13th, 2022, with a set budget of
 $4,300. We require accommodations that adhere to house rules regarding
visitors and should ideally be entire rooms. For food, we would love to
try a variety of cuisines, including Chinese, French, American, and
Mediterranean.

RFT Response:
Travel Plan:
Day 1:
Current City: from Reno to Abilene
Transportation: Self-driving, from Reno to Abilene, Duration: 22 hours 27
 mins, Distance: 2,412 km, Cost: 120
Breakfast: Thai Garden, Abilene
Attraction: The Grace Museum, Abilene
Lunch: Crispy Crust, Abilene
Dinner: Mx Corn, Abilene
Accommodation: Apartment minutes from manhattan, Abilene

Day 2:
Current City: Abilene
Transportation: -
Breakfast: Cakes Degree, Abilene
Attraction: Frontier Texas!, Abilene; Historic Fort Phantom Hill, Abilene
Lunch: LPK Waterfront, Abilene
Dinner: Pawan Foods, Abilene
Accommodation: Apartment minutes from manhattan, Abilene

Day 3:
Current City: from Abilene to Amarillo
Transportation: Self-driving, from Abilene to Amarillo, Duration: 4 hours
 10 mins, Distance: 446 km, Cost: 22
Breakfast: Wood Box Cafe, Amarillo
Attraction: Cadillac Ranch, Amarillo; Amarillo Botanical Gardens,
Amarillo
Lunch: Anand Restaurant, Amarillo
Dinner: Thalaivar, Amarillo
Accommodation: Trendy Brooklyn Room - 20mins from Manhattan, Amarillo

Day 4:
Current City: Amarillo
Transportation: -
Breakfast: The Cinnamon Kitchen, Amarillo
Attraction: Amarillo Zoo, Amarillo; Don Harrington Discovery Center,
Amarillo
Lunch: Punjabi Chaap Corner, Amarillo
Dinner: Cafe Coffee Day, Amarillo
Accommodation: Trendy Brooklyn Room - 20mins from Manhattan, Amarillo

Day 5:
Current City: from Amarillo to Lubbock
Transportation: Self-driving, from Amarillo to Lubbock, Duration: 1 hour
47 mins, Distance: 197 km, Cost: 9
Breakfast: Grand Barbeque Buffet Restaurant, Lubbock
Attraction: Buddy Holly Center, Lubbock; National Ranching Heritage
Center, Lubbock
Lunch: Kapoor's Sanjha Chulha, Lubbock
Dinner: San Carlo, Lubbock
Accommodation: Gorgeous Spacious Room in Clinton Hill, Lubbock
```

```
Day 6:
Current City: Lubbock
Transportation: -
Breakfast: Assam Tea Corner, Lubbock
Attraction: American Windmill Museum, Lubbock; Lubbock Lake National
Historic Landmark, Lubbock
Lunch: Domino's Pizza, Lubbock
Dinner: Platinum Lounge, Lubbock
Accommodation: Gorgeous Spacious Room in Clinton Hill, Lubbock

Day 7:
Current City: from Lubbock to Reno
Transportation: Self-driving, from Lubbock to Reno, Duration: 20 hours 3
mins, Distance: 2,145 km, Cost: 107
Breakfast: -
Attraction: -
Lunch: -
Dinner: -
Accommodation: -

CARL Response:
Travel Plan:
Day 1:
Current City: from Reno to Abilene
Transportation: Self-driving, from Reno to Abilene, duration: 22 hours 27
 mins, distance: 2,412 km, cost: 120
Breakfast: -
Attraction: -
Lunch: -
Dinner: Thai Garden, Abilene
(Average Cost:  $13, Cuisines:  Cafe, American, Desserts)
Accommodation: NYC Studio near Central Park and the Hudson River, Abilene
 (price: $304, room type: Entire home/apt, house rules: No pets, minimum
nights: 2)

Day 2:
Current City: Abilene
Transportation: -
Breakfast: Crispy Crust, Abilene (Average Cost: $54, Cuisines: Desserts,
Bakery, Cafe, American, Seafood)
Attraction: The Grace Museum, Abilene
Lunch: Mx Corn, Abilene
(Average Cost:  $62, Cuisines:  Tea, Cafe, Italian)
Dinner: The Grand Trunk Road, Abilene (Average Cost: $80, Cuisines: Tea,
Pizza, Bakery, BBQ, Chinese, Mediterranean)
Accommodation: NYC Studio near Central Park and the Hudson River, Abilene

Day 3:
Current City: from Abilene to Amarillo
Transportation: Self-driving, from Abilene to Amarillo, duration: 4 hours
 10 mins, distance: 446 km, cost: 22
Breakfast: -
Attraction: -
Lunch: -
Dinner: Wood Box Cafe, Amarillo (Average Cost: $11, Cuisines: Desserts,
French, Fast Food, Cafe, American)
Accommodation: Spacious retreat, Amarillo (price: $528, room type: Entire
 home/apt, house rules: No smoking, minimum nights: 2)

Day 4:
Current City: Amarillo
Transportation: -
Breakfast: Komachi, Amarillo (Average Cost: $71, Cuisines: Desserts, Tea,
 Fast Food, Chinese, Mediterranean)
```

```
Attraction: Cadillac Ranch, Amarillo
Lunch: Anand Restaurant, Amarillo (Average Cost: $53, Cuisines: Fast Food
, Seafood, Desserts, Italian)
Dinner: Shahi Food Corner, Amarillo (Average Cost: $90, Cuisines: French,
 Pizza, Bakery, Mediterranean)
Accommodation: Spacious retreat, Amarillo

Day 5:
Current City: from Amarillo to Lubbock
Transportation: Self-driving, from Amarillo to Lubbock, duration: 1 hour
47 mins, distance: 197 km, cost: 9
Breakfast: -
Attraction: -
Lunch: -
Dinner: Annapurna Bhandar, Lubbock (Average Cost: $78, Cuisines: French,
Chinese, Desserts, Fast Food)
Accommodation: Cozy Clean Small Apartment, Lubbock (price: $313, room
type: Entire home/apt, house rules: No children under 10 & No parties,
minimum nights: 2)

Day 6:
Current City: Lubbock
Transportation: -
Breakfast: Paris 6 Classique, Lubbock (Average Cost: $91, Cuisines:
Desserts, Cafe, Pizza, Mexican, BBQ, Chinese, Seafood)
Attraction: Buddy Holly Center, Lubbock
Lunch: Kapoor's Sanjha Chulha, Lubbock (Average Cost: $21, Cuisines: Tea,
 Cafe, Indian, American)
Dinner: Spezia Deliveries, Lubbock (Average Cost: $64, Cuisines: Desserts
, Italian, Bakery, Cafe, Mediterranean, Seafood)
Accommodation: Cozy Clean Small Apartment, Lubbock

Day 7:
Current City: from Lubbock to Reno
Transportation: Self-driving, from Lubbock to Reno, duration: 20 hours 3
mins, distance: 2,145 km, cost: 107
Breakfast: -
Attraction: -
Lunch: -
Dinner: -
Accommodation: -
```

## H THE USE OF LARGE LANGUAGE MODELS (LLMs)

In the preparation of this manuscript, we utilized a Large Language Model (LLM) as an assistive tool for language polishing. The primary role of the LLM was to improve grammar, refine sentence structure for clarity, and ensure a consistent and professional tone. We confirm that all scientific contributions, including research ideation, experimental design, data analysis, and the core writing of the content, were exclusively the work of the human authors. The LLM's function was limited to that of a writing assistant and did not extend to any aspect of the intellectual contribution to this work.

