# OpenReview forum: "CARL: Constraint-Aware Reinforcement Learning for Planning with LLMs"
_ICLR.cc/2026/Conference — ICLR 2026 Conference Withdrawn Submission_

### Official Review · Reviewer_RpQe · 2025-10-24

**Soundness:** 3
**Presentation:** 4
**Contribution:** 2
**Rating:** 2
**Confidence:** 4

**Summary:**

This paper proposes CARL (Constraint-Aware Reinforcement Learning), a training framework to improve Large Language Models’ (LLMs) adherence to task constraints in planning problems. CARL leverages reinforcement learning to enhance a model’s intrinsic sensitivity to constraints by comparing its responses under constrained and unconstrained inputs. This comparison is used to shape/construct constraint-aware reward/preference that increases the attribution score of constraint-related tokens, thereby promoting constraint-consistent planning behavior. The approach integrates with both on-policy and off-policy reinforcement learning algorithms, including GRPO and DPO. The authors evaluate CARL on three benchmarks: BlocksWorld (symbolic planning), TravelPlanner (real-world travel planning), and T-Eval (tool-use reasoning). The also compare it against Reinforcement Fine-Tuning (RFT) and large reasoning models such as o1-preview and DeepSeek-R1. Additionally, the paper provides comparisons on other planning benchmarks, namely Mystery BlocksWorld and TripCraft, to assess the generalizability of the proposed approach.

**Strengths:**

The paper addresses the challenging combinatorial problem of planning under constraints using Large Language Models (LLMs). It is clearly written and easy to follow. The proposed framework accommodates both explicit and implicit constraint types, although the latter requires auxiliary extraction tools. One of the core element of the method is a novel reward formulation based on the KL divergence between constrained and unconstrained policy distributions, enabling training under on-policy reinforcement learning. The second core element of the method is the strategic preference construction approach for off-policy settings, pairing constrained and unconstrained responses to guide DPO-style updates. These ideas are intuitive, interpretable, and mathematically sound.

Empirically, CARL is evaluated with both on-policy (GRPO) and off-policy (DPO) algorithms across three distinct planning benchmarks, BlocksWorld, TravelPlanner, and T-Eval. The results demonstrate consistent and meaningful improvements over Reinforcement Fine-Tuning (RFT) baselines, achieving up to a +11.1% gain on the TravelPlanner benchmark, which is a notably challenging environment with multiple interacting constraints. Importantly, CARL-trained models also outperform large reasoning models such as o1-preview and DeepSeek-R1 on some of the trained benchmarks. The attribution analysis (Fig. 5) further validates the mechanism, showing that CARL increases the model’s focus on constraint-relevant tokens in alignment with improved task success rates, although not conclusively for the challenging TravelPlanner benchmark. Overall, the paper’s strengths lie in its conceptual simplicity and general applicability across RL frameworks.

**Weaknesses:**

The work has the following weaknesses, which limit its effectiveness, rigor, and generalizability.

1. **Reward Shaping Without Semantic Guarantee.** The proposed KL-based reward measures the distributional shift between constrained and unconstrained outputs, but this signal is not necessarily aligned with constraint satisfaction. A high KL divergence can indicate that the model’s response changes when a constraint is added, yet the change may lead to an invalid plan. This introduces the possibility of reward hacking, in which models exaggerate words related to constraints or hallucinate irrelevant content to increase KL. Indeed, Appendix B (Fig. 6) shows that training without KL penalties leads to collapse, confirming instability. Thus, the reward formulation is heuristic rather than theoretically justified, and its connection to constraint correctness remains unproven.

2. **Dependence of Preference Optimization on Base Model Competence.** The off-policy variant (CA-DPO) assumes that the base model can already generate at least one valid constrained response per input, so that the “constrained” response is meaningfully better than the “unconstrained” response. However, both are based on the same base model, differing only in prompt conditioning. If the base model violates constraints in both cases, the preference label becomes meaningless, and the optimization may merely reinforce lexical differences rather than achieve true compliance. This circular dependency severely limits CARL’s effectiveness for weaker or misaligned base models. A robust implementation would require an external correctness oracle or a constraint verifier, but neither is provided.

3. **Dependence on External Constraint Extraction Tools.** The framework relies on external or heuristic methods to identify constraint tokens, particularly in implicit constraint settings (e.g., T-Eval), where the authors use GPT-4o to extract constraints. At present, there is no ablation analysis provided on the accuracy or quality of the extraction tool or its impact on the algorithm’s performance. If constraint identification is noisy, the entire training signal becomes unreliable.

4. **Weak Generalization and Scalability.** The paper’s claims about generalization and scalability are not convincingly backed by empirical evidence. The generalization results in Appendix D are notably poor. E.g., less than 2\% success on out-of-distribution variants of Mystery BlocksWorld and TripCraf undermines the claim of general-purpose constraint reasoning and raises concerns about the method's merit.

5. **Misleading Comparisons with Large Reasoning Models.** The baselines are evaluated in zero-shot settings, while CARL is task-specifically fine-tuned. This makes the comparison somewhat misleading. The only fair comparison is against Reinforcement Fine-Tuning (RFT), where CARL indeed shows measurable gains (e.g., +11.1% on TravelPlanner).

**Questions:**

1. How does the method perform with various extraction tools, and how does this affect performance?
2. Why does the method not generalize well to TripCraft and Mystery BlocksWorld?
3. Are there any theoretical justifications or existing results that underpin the idea of why it should generalize to other tasks in a zero-shot setting?
4. How well does the method scale in challenging combinatorial planning tasks under constraints, e.g., multi-robot collaborative tasks?
5. Why is there no significant change noticed in the attribution score (Fig.~5) for TravelPlanner using CARL? What is the reason for this, and does it explain the unexpectedly poor generalization performance?

---

### Official Review · Reviewer_wzXm · 2025-10-31

**Soundness:** 2
**Presentation:** 3
**Contribution:** 3
**Rating:** 4
**Confidence:** 3

**Summary:**

This paper introduces CARL (Constraint-Aware Reinforcement Learning), a training-based framework designed to enhance the intrinsic ability of Large Language Models (LLMs) to adhere to constraints during planning tasks. CARL introduces a constraint-aware reward signal that compares model output distributions under constrained versus unconstrained inputs using KL divergence, thus encouraging constraint-focused behavior. The authors evaluate CARL on three planning benchmarks (BlocksWorld, TravelPlanner, and T-Eval) and demonstrate improvements over standard reinforcement fine-tuning (RFT) and state-of-the-art reasoning models. The method is shown to be compatible with both on-policy (GRPO, PPO) and off-policy (DPO) RL algorithms.

**Strengths:**

* The paper proposes to mitigate the problem of constraint ignorance or violation, which is a clear and important weakness in current LLMs for planning. They propose to use measure constraint sensitivity through distributional shifts between constrained and unconstrained inputs as a reward signal, which is novel and sound.
* The framework is evaluated across three very distinct benchmarks with different constraint types, which demonstrates the generality. The empirical results are strong and analysis are comprehensive. CARL achieves consistent and substantial performance improvements across three benchmarks.
* The authors show that the method is compatible with both on-policy (GRPO, PPO) and off-policy (DPO) RL algorithms

**Weaknesses:**

Some details of the framework are missing. The procedure of getting constraints from the query is not clear. In addition, the inputs/outputs for proposed method and baselines are ambiguous. More analysis is needed. See questions for detail.

**Questions:**

* First, as stated in the paper, for tasks with explicitly stated constraints, CARL directly use the constraints as described in the query. However, how are the commonsense constraints of TravelPlanner extracted/defined as they are not included in the query? Do you assume access to descriptions of all implicit/commonsense constraints? If so, do the baselines also have access to the list of constraints they need to care about? If not, what would be the performance if they have?
* Then, for tasks with implicitly defined constraints such as T-Eval, the framework uses LLM to extract constraints from the query. However, the performance of this extraction is not tested or discussed. Are there any failure cases? How could the performance degrade if this extraction quality gets worse for more complex questions? What happens when constraints are misidentified or partially extracted? Current extraction is based on a pre-defined prompt. How is this generalizable to other domains? Similarly, what would be the performance if the extracted constraints are emphasized in the baselines’ prompts?
* It is surprising to see the large performance gain with only 100, 45, and 128 training samples for the three benchmarks, especially when the queries of some benchmarks are very complex. Can you add some analysis of the underlying reasons?

---

### Official Review · Reviewer_zFw1 · 2025-10-31

**Soundness:** 2
**Presentation:** 2
**Contribution:** 2
**Rating:** 4
**Confidence:** 4

**Summary:**

This paper addresses the issue of Large Language Models (LLMs) tending to overlook constraints during planning by decoupling queries into goals and constraints. During training, constraints are specifically optimized using two types of rewards: task-oriented objectives and constraint-sensitive adherence. Experimental results and ablation studies demonstrate the effectiveness of the proposed approach.

**Strengths:**

1. The paper identifies a valid and important issue: the tendency of large language models (LLMs) to overlook constraints. This is a meaningful problem that deserves attention.
2. The proposed approach is simple yet effective, leveraging reinforcement learning (RL) to optimize adherence to constraints. The use of task-oriented objectives and constraint-sensitive rewards is well-motivated.
3. The experimental results, including ablation studies, are comprehensive and sufficiently demonstrate the effectiveness of the proposed method on Qwen3-8B.

**Weaknesses:**

1. The paper adopts Qwen3-8B as the base model, which is a relatively weaker LLM compared to state-of-the-art models. While the approach is validated on this model, the results may not generalize to larger or more powerful LLMs.
2. The issue of constraints being overlooked by LLMs might naturally be mitigated with larger models and more diverse pretraining datasets. While the paper demonstrates improvement through targeted training, its practical significance could be questioned if scaling up the model can solve the problem without additional effort. This point should have been discussed or compared more thoroughly.
3. The method appears to enhance planning-related tasks, but it is unclear whether it might degrade performance on other tasks. The lack of broader experimental results limits the understanding of potential trade-offs.
4. The novelty of the proposed method is somewhat limited. The approach to optimizing constraints during training could be applied to minor issues in any LLM, and it does not provide a deeper theoretical insight into why LLMs overlook constraints. For example, the paper does not explore whether this issue arises due to contradictions or biases in pretraining data, nor does it discuss potential opportunities to address the problem during pretraining.

**Questions:**

1. Have you tested the proposed method on stronger LLMs or compared its effectiveness across models with different scales? For example, how does the performance change when using a larger or more advanced model like GPT-4 or LLaMA-2?
2. Did you compare the proposed approach to simply scaling up the pretraining data or model size? If larger models naturally improve constraint adherence, how does your method justify its practical utility?
3. What is the computational overhead introduced by the reinforcement learning approach? Is the method scalable when applied to much larger models?

---

### Note · Authors · 2025-11-21

I have read and agree with the venue's withdrawal policy on behalf of myself and my co-authors.